# Initial pulse of Siberian Traps sills as the trigger of the end-Permian mass extinction

S.D. Burgess[1], J.D. Muirhead[2] & S.A. Bowring[3]

Mass extinction events are short-lived and characterized by catastrophic biosphere collapse and subsequent reorganization. Their abrupt nature necessitates a similarly short-lived trigger, and large igneous province magmatism is often implicated. However, large igneous provinces are long-lived compared to mass extinctions. Therefore, if large igneous provinces are an effective trigger, a subinterval of magmatism must be responsible for driving deleterious environmental effects. The onset of Earth's most severe extinction, the end-Permian, coincided with an abrupt change in the emplacement style of the contemporaneous Siberian Traps large igneous province, from dominantly flood lavas to sill intrusions. Here we identify the initial emplacement pulse of laterally extensive sills as the critical deadly interval. Heat from these sills exposed untapped volatile-fertile sediments to contact metamorphism, likely liberating the massive greenhouse gas volumes needed to drive extinction. These observations suggest that large igneous provinces characterized by sill complexes are more likely to trigger catastrophic global environmental change than their flood basalt- and/or dike-dominated counterparts.

[1] U.S. Geological Survey, Volcano Science Center, 345 Middlefield Road, Mail Stop 910, Menlo Park, CA 94025, USA. [2] Department of Earth Sciences Syracuse University, 204 Heroy Geology Laboratory, Syracuse, NY 13244, USA. [3] Earth, Atmospheric, and Planetary Sciences Department Massachusetts Institute of Technology, 77 Massachusetts Avenue, Cambridge, MA 02139, USA. Correspondence and requests for materials should be addressed to S.D.B. (email: sburgess@usgs.gov)

Large igneous province (LIP) magmatism[1] and related greenhouse gas emissions are implicated as the primary trigger for three of the five major Phanerozoic biotic crises, of which the end-Permian event was the most biologically severe, marking a critical inflection point in the evolutionary trajectory of life on Earth[2–4]. Although other triggers for the end-Permian event have been proposed[5], a causal connection between Siberian Traps LIP magmatism and this mass extinction is favored. This causal connection is supported by evidence for a striking temporal coincidence between the two phenomena[4, 6–8], rapid introduction of isotopically light carbon into the marine system[8, 9], an abrupt increase in global sea surface temperature (~ 10°C)[10], and the physiological selectivity of marine extinction patterns[11]. These lines of evidence point unequivocally toward a massive, short-lived input of greenhouse gasses (e.g., $CO_2$, $CH_4$) to the atmosphere, which is thought to have been generated in sufficient quantity either by contact metamorphism of crustal sediments during Siberian Traps LIP magma emplacement[12, 13] or during LIP plume-related melting at the base of the lithosphere[14].

Three issues complicate the proposed causal linkages between mass extinction and LIP magmatism. The first is a significant disparity in the timescales over which LIP magmatism and mass extinction occur; magmatism lasts on the order of 1–5 Myr, with multi-pulsed examples lasting up to 50 Myr[1], whereas mass extinction happens on the order of <100 kyr[4, 8, 15–18] (Fig. 1). The second is the relative timing of LIP emplacement and mass extinction. Rather than both events coinciding at onset, in some cases, LIP emplacement began hundreds of thousands of years prior to mass extinction, with little to no discernable negative feedback in the biosphere during voluminous eruptions[4, 15, 19]. Third, not all LIP events are associated with marked environmental change[20, 21]. Given that LIPs are composed of multiple igneous components (e.g., pyroclastic rocks, lavas, dikes, and sills), which are often emplaced at different times and over varying intervals, one must critically assess which aliquant of the total magmatic volume, if any, drove biosphere collapse? And further, what trait of this aliquant distinguishes it from the remaining magmatic volume? The responsible volume must be demonstrably emplaced immediately prior to and possibly during the mass extinction, and must have the capacity to trigger massive greenhouse gas release. Recent geochronology on Siberian Traps LIP magmatism and the end-Permian extinction[4, 8] has highlighted a distinct temporal association between these two phenomena, but without uniquely identifying the specific extinction-triggering magma volume. This temporal framework nonetheless permits a detailed evaluation of the purported causal connection between magmatism and extinction to determine which subinterval of Siberian Traps LIP magma, if any, induced the mass extinction, and why this particular magma was so deadly.

Here we identify the deadly subinterval of Siberian Traps LIP magmatism as the initial pulse of sill emplacement into the volatile-fertile Tunguska basin. By considering these results in context with major extinction–LIP couplets within the past 300 Myr, we illustrate the importance of initial widespread LIP sill emplacement into volatile-fertile sediments for driving environmental change on a global scale.

## Results

**A relative timing framework.** Recent high-resolution U/Pb geochronology provides a detailed chronology of Siberian Traps magmatism and the end-Permian mass extinction[4, 8], and the ability to directly compare the relative timing of the two events. Building on the schema presented in ref. [4], we construct a framework to guide identification of the causative LIP subinterval. In this framework, the Siberian Traps magmatic activity can be segmented into three distinct emplacement stages. Stage 1, beginning just prior to 252.24 ± 0.1 Ma, was characterized by initial pyroclastic eruptions followed by lava effusion (Fig. 2). During this stage, an estimated 2/3 of the total volume of Siberian Traps lavas was emplaced (>1 × 10^6 km^3). Stage 2 began at 251.907 ± 0.067 Ma, and was characterized by cessation of extrusion and the onset of widespread sill-complex formation[4]. These sills are exposed over a >1.5 × 10^6 km^2 area, and form arguably the most aerially extensive continental sill complex on Earth. Intrusive magmatism continued throughout stage 2 with no apparent hiatus. Stage 2 ended at 251.483 ± 0.088 Ma, when extrusion of lavas resumed after an ~420 ka hiatus, marking the beginning of stage 3. Both extrusive and intrusive magmatism continued during stage 3, which lasted until at least 251.354 ± 0.088 Ma, an age defined by the youngest sill dated in the province[4]. A maximum date for the end of stage 3 is estimated at 250.2 ± 0.3 Ma[6].

Integration of LIP stages with the record of mass extinction and carbon cycle at the Permian-Triassic Global Stratotype Section and Point (GSSP)[4] shows three notable relationships. (1) Extrusive eruption during stage 1 of Siberian LIP magmatism occurs over the ~300 kyr prior to the onset of mass extinction at 251.941 ± 0.037 Ma[4, 8]. During this interval, the biosphere and the carbon cycle show little evidence of instability, although high-latitude environmental stress prior to the mass extinction has been observed[22] (Fig 2). (2) The onset of stage 2, marked by the oldest Siberian Traps sill[4], and cessation of lava extrusion, coincides with the onset of mass extinction and the abrupt (2–18 kyr) negative $\delta^{13}C$ excursion immediately preceding the extinction event[4]. The remainder of LIP stage 2, which is characterized by continued sill emplacement, coincides with broadly declining $\delta^{13}C$ values following the mass extinction. (3) Stage 3 in the LIP begins at the inflection point in $\delta^{13}C$ composition, whereupon the reservoir trends positive, toward pre-extinction values.

**Identifying the smoking gun.** The disparity in duration between Siberian LIP magmatism and the end-Permian mass extinction, and the observation that pre-extinction eruption of an estimated 2/3 of LIP lavas resulted in limited deleterious global forcing on the biosphere[23], suggests that only a restricted subinterval of LIP magmatism triggered the mass extinction. Siberian Traps lava

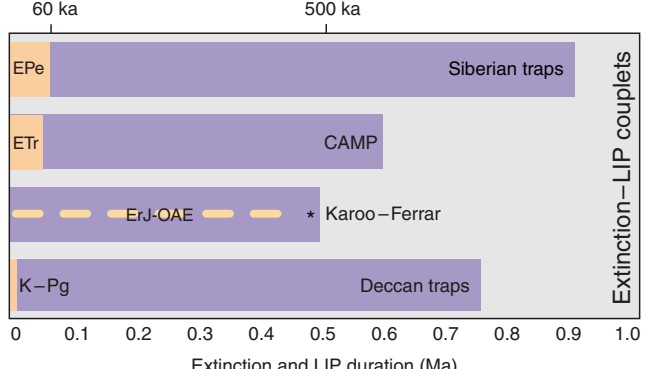

**Fig. 1** Temporal disparity between approximate durations of mass extinction and LIP events. Each couplet represents a temporally associated extinction and magmatic event, highlighting the prolonged duration of magmatism relative to punctuated mass extinction. Onset of coupled events is not necessarily contemporaneous. Durations from refs. [4, 8, 15–17, 34, 37, 38]. *CAMP* Central Atlantic magmatic province, *EPe* end-Permian extinction, *ETr* end-Triassic extinction, *ErJe OAE* early Jurassic ocean anoxic event, *K-Pg* Cretaceous Paleogene extinction

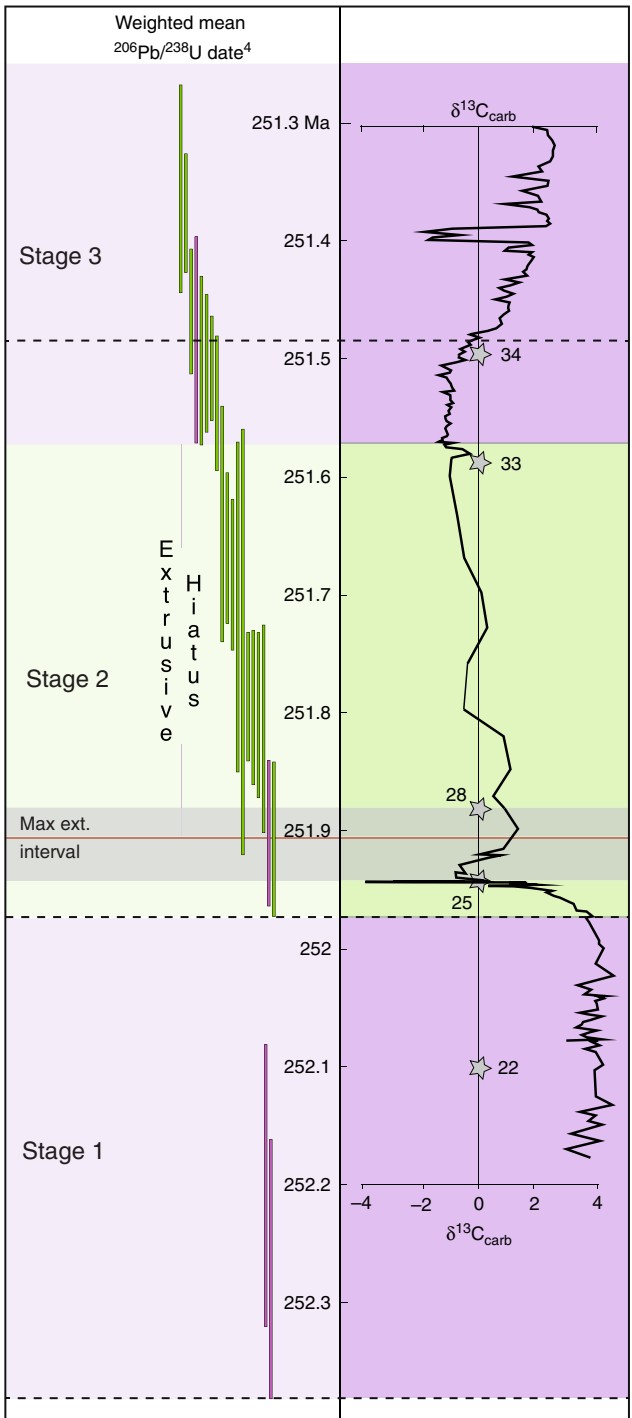

**Fig. 2** Stages of Siberian Traps LIP magmatism relative to timing of mass extinction. Stages are color coded to the dominant style of magmatism at the time, *purple* for extrusive, and *green* for intrusive. Weighted mean $^{206}Pb/^{238}U$ zircon dates are shown at 2-sigma, and are also color coded *purple* for extrusive and *green* for intrusive[4]. Carbonate carbon record from refs. [8, 38]. *Stars* located on the carbonate carbon record are stratigraphic locations of U/Pb dates[8]

higher volatile yields at depth than those released during accompanying LIP lava eruptions[14]. However, our preferred extinction age model[4] indicates that the end-Permian event did not occur during this period of theoretically higher plume-related devolatization (inferred to have initiated at 252.8 Ma[14]), but postdated it by ~900 ka.

A striking temporal coincidence is instead observed between the extinction event and the emplacement of Siberian Traps sills, which intruded the thick Tunguska basin, composed of evaporite, clastic, carbonate, and hydrocarbon-bearing rocks[12]. Heating of sediments over the large area encompassed by the sill complex ($>1.5 \times 10^6$ km$^2$) likely liberated massive volumes of greenhouse gasses[12, 25]. However, sills were intruded over an interval of ~500 kyr[4], while the extinction interval and carbon isotope anomaly are both an order of magnitude shorter. This disparity necessitates that a currently unidentified subinterval of intrusive magmatism during stage 2 is the extinction-triggering aliquant. The oldest sills dated in the magmatic province mark the onset of stage 2[4], and are the only sills whose emplacement timing is coincident with both the carbon isotope excursion and the onset of mass extinction; all other dated sills postdate these events (Fig. 2), and are thus disqualified as the trigger. Therefore, we suggest that only intrusions emplaced at the beginning of stage 2, during the initial lateral growth of the Siberian Traps sill complex, satisfy all the necessary criteria to qualify as the trigger of the end-Permian mass extinction.

The exact cause for the observed transition from lava eruptions to widespread sill complex formation during stage 2 is challenging to pinpoint. However, emplacement of these sills occurred immediately after cessation of stage 1 lava extrusion, possibly in response to or as a result of construction of a thick volcanic load atop the lithosphere during stage 1 magmatism (Fig. 3). Numerical modeling studies illustrate that volcanic loads can impose a compressive stress state in the underlying crust that impedes magma ascent[26], and in some instances, will promote sill formation[27]. Transitions from dominantly sub-vertical lava feeder systems to horizontal modes of intrusion (i.e., sills) are also attributed to rotation of the least compressive stress direction due to high magma flux into the feeder network[28], and contrasting mechanical properties between layers of varying density and rigidity[29]. Once the Siberian sill complex was established, sill intrusion continued into stage 3, coincident with renewed lava effusion. The impetus for stage 3 extrusive magmatism in the northern region of the LIP is unclear, but may be the result of a new eruption center[30], or eclipsing of a critical magmatic overpressure threshold at depth[26].

Prior to growth of the sill complex, magmas feeding stage 1 lavas likely transited the crust rapidly through sub-vertical conduits within a narrow area presently buried either beneath the Siberian LIP lavas, or represented as sparsely exposed (possibly radiating) dikes in northern parts of the province[30]. Consequently, only a small volume of basin sediments experienced contact metamorphism during stage 1 and thus environmental impacts were likely relatively negligible. Not until the Siberian LIP emplacement style changed at the onset of stage 2 from being dominantly extrusive to intrusive was the "volatile-fertile" Tunguska basin subjected to widespread contact metamorphism, thereby maximizing the likelihood of heat transfer and massive volatile yield (Fig. 3)[12].

Lateral sill-complex emplacement continued, subsequent to the initial pulse, but into a basin depleted of its initially high volatile content. Heat transfer to sediments from these later sills still generated greenhouse gasses; their effects are evident from the prolonged isotopically negative $\delta^{13}C$ interval at the GSSP, and the sustained elevation in global sea surface temperature[8, 10]. The biosphere's ability to buffer itself against prolonged, rather than

eruptions appear to have ceased immediately prior to the extinction, and these lavas likely lack the dissolved $CO_2$ contents required to drive a significant global heating event[24], both of which suggest the causative aliquot was probably not extrusive. Initial impingement of a plume head on the lithosphere, and related melting, is also hypothesized to generate significantly

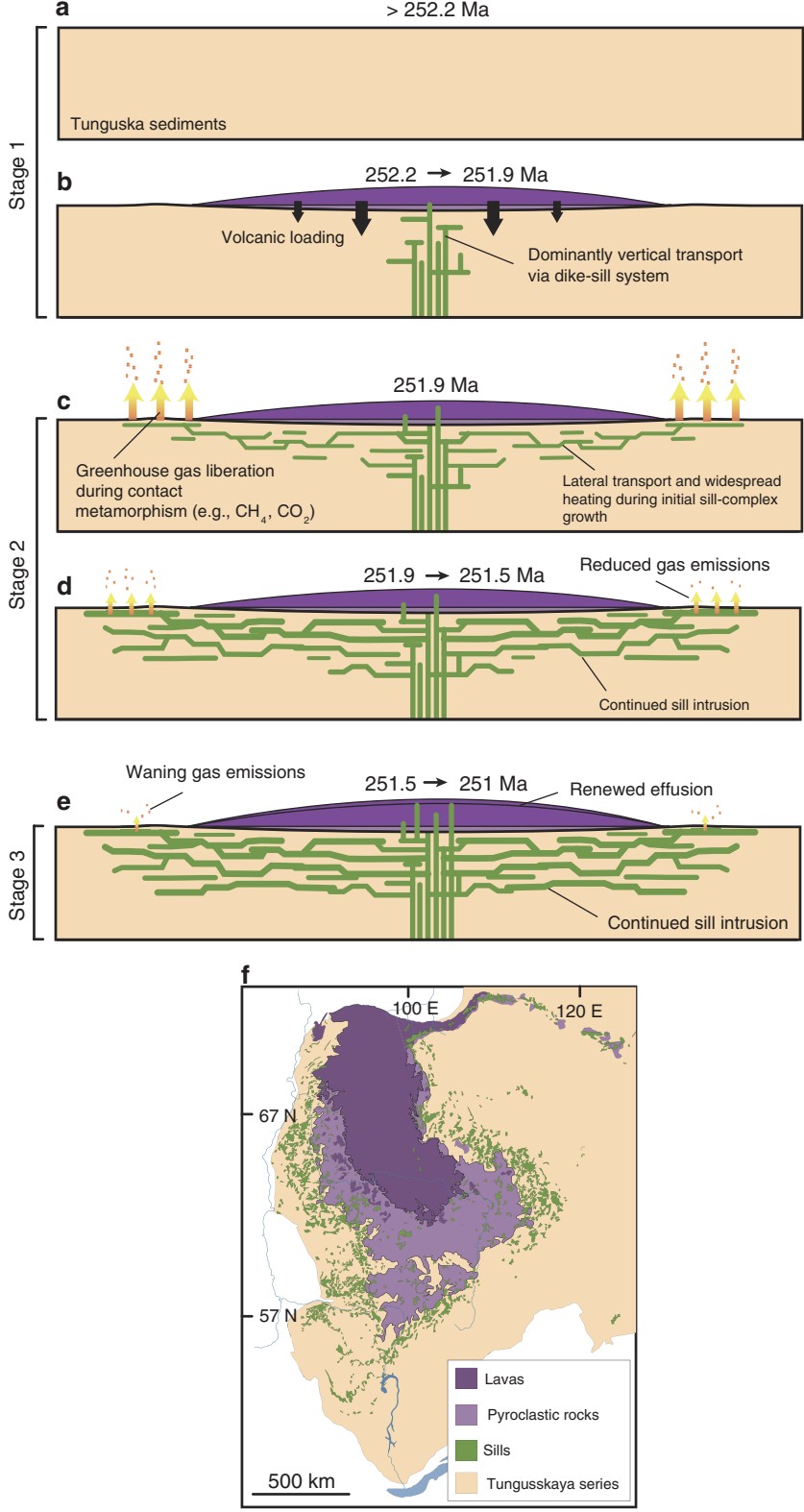

**Fig. 3** Time series of Siberian Traps LIP emplacement. **a** Pre-emplacement basin. **b** Emplacement of a volcanic load during stage 1. The feeder system is unresolved, and most likely situated below lavas in **f**. **c** Beginning of stage 2, with lateral sill complex growth, widespread heating, and greenhouse gas generation. **d** Continued sill emplacement during stage 2. **e** Renewed extrusive magmatism during stage 3. Geochronology defining time-steps from ref. [4]. Map inset **f** modified from ref. [12]

initially large and abrupt greenhouse gas input, however, likely mitigated many of the deleterious effects of this gas generation[24]. Sluggish biotic recovery following extinction may well be attributed to stages 2 and 3 magmatism, but the rapid loss in global biodiversity characterizing the mass extinction suggests an equally rapid trigger. Initial intrusion of the Siberian Traps sill complex represents such a trigger, as it was short lived, coincident with the mass extinction, and capable of producing the large amount of climate-altering gasses required to drive biosphere collapse.

**Making a deadly LIP**. Our model suggests that LIPs characterized by sill-complex formation are more likely to trigger mass extinction than their flood basalt- and/or dike-dominated counterparts. The composition of sediment into which sills are emplaced is also critically important[31, 32], as a sill network built within a volatile-poor substrate will not result in volatile generation on the scale necessary to drive extinction. Some of the largest LIP-related extinction events (>13% genus-level extinction[33]) in the last 300 Myr are demonstrably associated with widespread sill emplacement into sedimentary basins during Siberian Traps, Central Atlantic magmatic province, and Karoo-Ferrar LIP magmatism[4, 15, 17] (Fig. 1). The notable exception over this period is the flood basalt-/dike-dominated Deccan Traps LIP, which is temporally associated with the ~66 Ma Cretaceous-Paleogene (K-Pg) extinction event[19, 34]. The model presented here suggests a limited role for Deccan magmatism in triggering the K-Pg event, which would shift burden to the contemporaneous Chicxulub impact[34]. It is plausible that the Deccan LIP and Chicxulub impact shared the causal burden[34], and it is therefore possible that neither would have driven extinction of K-Pg magnitude acting alone[35]. The model presented here suggests that LIPs characterized by a pulse of widespread sill emplacement into a volatile-fertile basin are lethal on a global scale.

**Data availability**. Geochronology supporting the model presented here is available through refs. [4, 8]. Treatment of this data is available from the corresponding author, S.D.B.

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

## Acknowledgements

The authors would like to acknowledge J.T. Hagstrum and B. Schoene for thoughtful comments, and A.R. Van Eaton, A.T. Calvert, M.A. Coble, D.T. Downs, J.A. Vazquez, and C.R. Bacon for discussion during development of this manuscript. S.D.B. would like to acknowledge the USGS Mendenhall postdoctoral program.

## Author contributions

S.D.B. wrote the manuscript. J.D.M. and S.A.B. contributed intellectual and editorial advisement.

## Additional information

**Competing interests:** The authors declare no competing financial interests.

