## [Peer Review File · Nature Communications]

Reviewers' Comments:

Reviewer #1 (Remarks to the Author)

Review by Richard Ernst of Burgess, Muirhead and Bowring "Initial Siberian Traps sill emplacement as the trigger of the end-Permian mass extinction"

GENERAL COMMENTS

This is another breakthrough publication by Seth Burgess and colleagues. I heard (and was impressed by) their talk at the annual GSA meeting (Sept 2016) on the theme that has been developed into this full paper. Glad to see these new insights developed into a full paper. Previously the high precision U-Pb dating by Burgess and Bowring (Science Advances) on about 25 samples from the Siberian trap event demonstrated incredibly short duration of the LIP event and the overlap with the mass extinction. At the time it was stunning that the Siberian Trap event was shown to have a duration of less than 1 myr. With this current manuscript (submitted for Nature Communications) Seth Burgess has taken the data interpretation to a new level of being able to link the extinction to a specific point in the overall (short) 1 myr duration. The notion that the extinction was linked to the start of the intrusive-only phase of the Siberian trap LIP is consistent with the emerging evidence from the Oslo group (e.g Svensen et al. 2009) for the importance (even dominance) in many LIPs of gas release from interaction of sills with volatile-rich host rocks and transport of these gases to the atmosphere through Hydrothermal Vent Complexes.

I suggest publication with minor revisions. However, a few minor points below need to be addressed

DETAILED COMMENTS

It is too bad that there are NOT line numbers as this would facilitate more easy commentary.

ABSTRACT: "However, LIP magmatism is long-lived relative to extinction.." In this new 'universe' that Burgess and other high precision U-Pb daters are bringing us to— a less than one myr duration for a LIP event is becoming normal (rather than shockingly short). However, there are also some LIP events that in fact are much longer and consist of multiple short pulses that potentially span 10s of myr-- e.g. 1115-1085 Ma Keweenaw LIP of North America (Ernst 2014).

INTRODUCTION:

Could I suggest referencing the only text book on LIPs (Ernst, R.E. 2014. Large Igneous Provinces. Cambridge University Press, 653 p.)—

and perhaps adding this citation just after LIPs are first mentioned: "Large igneous province (LIP) magmatism" (Ernst, 2014)

"The first is a significant disparity in the timescales over which LIP magmatism and mass extinctions occur" It is again so noteworthy that Burgess (and other geochronologists) have shown us that a 1 million year duration LIP event is in fact LONG compared to the duration of a mass extinction. What an exciting new world of LIP geochronology we are in now!

In the Introduction: "Thirdly, not all LIP events are associated with marked environmental change" and the reference used here is Stothers 1993. Can a more recent reference also be added: e.g.

Bond, David P.G., Grasby, Stephen E., On the causes of mass extinctions, *Palaeogeography, Palaeoclimatology, Palaeoecology* (2016), doi: 10.1016/j.palaeo.2016.11.005

Or Courtillot, V.E., Renne, P.R., 2003. On the ages of flood basalt events. *Comptes Rendus Geosciences* 335, 113–140

[I have a paper explicitly on this theme of linking of LIP events and environmental impacts for PPP,

but unfortunately it is a few weeks away from being accepted.]

DISCUSSION section, third paragraph:

You are proposing that Stage 1 lavas represent a “structural barrier that inhibit further magma ascent to the surface”. This would imply that the first sills should be emplaced right below the flood basalt package and that the sill complex should progressively build downward (i.e. deeper and deeper in the crust). This seems to be implied in Figure 3. An interesting question is what thickness the flood basalt package needs to attain before the flood basalt package begins to act a “structural barrier”--

“crustal loading / unloading” model: I am not understanding this model. I can see the potential impact of erosion to cause unloading, leading to sill emplacement. Is this what you have in mind? Or are you producing the sills during the crustal loading stage—I don’t see how

Does the chemistry of Stage 1 magmatism differ from Stage 2 magmatism? I would think that the initial lavas which ascend directly to the surface are less likely to exhibit any crustal chemical contamination. However, the Stage 2 intrusives will have more significant interactions with their host rocks (to produce the gases associated with extinction) and should potentially be more crustally contaminated. Is there any geochemistry data that compares the flood basalts of Stage 1 with the intrusives of Stage 2. In this regard it is interesting that the Nadezhdinsky lavas (part of Stage 1) do exhibit crustal contamination. So it would seem that at least some Stage 1 lavas still did pause and interact with continental crust, before ascending to be emplaced as flows.

At the end of this paragraph you note “the striking outcrop pattern... which is characterized by extrusive rocks in the interior of the province and a zone of younger sills flanking these rocks around nearly the entire periphery of the province” You consider this to be a primary pattern, but I wonder if this pattern could simply represent deeper erosion (and removal of the flood basalts) at the periphery of the province.

DISCUSSION section, fourth paragraph:

In your discussion of the location of the feeder system to the Stage 1 lavas, you could note that we have identified a giant radiating and giant circumferential swarm that both locate a mantle plume centre just to the north of the main volcanics, at a location of approximately 100 E, 72 N (Ernst and Buchan, 2001 GSA Special Paper 352, p. 247-265; see also Figure 16.4 in Ernst, 2014)

DISCUSSION section

An additional consideration is that if the accumulation of Stage 1 lavas represents a structural barrier that leads to a stage 2 dominated by intrusions, then what changes to allow lava flows to be again emplaced at the following stage 3?

FIGURE CAPTIONS

Figure 1 Caption: in the caption you need add a link to the colours, blue and red. So after “mass extinction” add “(blue)” and after “large igneous province (LIP)” add “red”. In the caption explain: EPE, ETE, ErJE, OAE and ECE. The key reference for the U-Pb dating for the Karoo portion of the Karoo-Ferrar LIP is missing: Add this reference: Svensen, H., Corfu, F., Polteau, S., Hammer, Ø., Planke, S., 2012. Rapid magma em-placement in the Karoo Large Igneous Province. Earth Planet. Sci. Lett.325–326, 1–9. doi 10.1016/j.epsl.2012.01.015.

Figure 2 caption: explain the stars in the carbon isotope record

FIGURE 1: Very powerful diagram

FIGURE 3: this model assumes no erosion of flood basalts from the periphery of the province (where the sills are dominant—see discussion above). However, we know that major erosion occurred in the vicinity of the Anabar shield. A mantle plume centre could be added to the diagram (based on Ernst and Buchan 2001 or in Ernst 2014) at a location of approximately 100 E, 72 N

Reviewer #2 (Remarks to the Author)

Review of Burgess et al: Initial Siberian Traps sill emplacement as the trigger of the end-Permian mass extinction

This manuscript offers a new and insightful hypothesis for the volcanogenic mechanism that drove the end-Permian extinction. The hypothesis is novel, important for the understanding of the relationship between large igneous provinces and mass extinction events, anchored firmly in the state-of-the-art dataset now available, and is therefore worthy of consideration for this journal. I have a few issues with the manuscript in its current form, which may affect the final publication of the manuscript.

Major Points:

1) While the observation that the first pulse of sill emplacements into the Tunguska basin is coincident with the end-Permian mass extinction is new and extremely important, this paper does not currently have any 'new' results. The current Results Section is a description of the work the authors and others have done previously. I don't think this negates this from being published in Nature Communications, but I do feel that it needs to be restructured somewhat to reflect that this is a new hypothesis to explain existing data.

2) The argument put forward is based on the remnant outcrops of the Siberian Traps in the Tunguska Basin. There needs to be a few sentences added to address the certainty that the authors feel they have that the whole story is covered by the surviving exposures.

3) Discussion: This section makes the assumption that LIP lavas are too low in CO₂ to be a primary source of carbon. Calculated CO₂ volumes of individual lavas may not be representative of total magmatic degassing from LIPs. For example, the arrival of a plume head may be considerably richer in volatiles compared to lavas that came later in the LIP emplacement (this doesn't fit the data presented in this paper, but needs to be discussed nonetheless). Moreover, the thermal alteration of the lithosphere as the plume head arrived beneath the crust may have led to considerable degassing of CO₂ (e.g. Sobolev et al, Nature, 2011). Can the authors be certain that this slightly delayed pulse isn't due to this volatile rich front reaching the surface? I understand that this paper is proposing an alternative hypothesis, but the text needs to be changed to reflect that this is one of several possible avenues that could explain the carbon isotope excursion.

Minor Points:

Abstract: 'Sub-interval' instead of 'sub interval'.

Introduction, Paragraph 1: References 1 and 2 are out of order

Introduction, Paragraph 1: The greenhouse gas H₂O is released in negligible quantities compared to surface reservoirs, and will not affect the radiative properties of the Earth as a primary driver, only as a positive feedback to other greenhouse gas emissions (see the review of Jones et al, 2016 and references therein).

Introduction, Paragraph 2: There is an implicit assumption in this writing that the mass extinction can only be caused by volcanism. It is highly likely that it was, but I think a degree of caution with the language used here would be prudent.

Discussion, Paragraph 1: End sentence, recommend changing to ‘...qualifying aliquot is dominated by intrusive sources.’

Discussion, Paragraph 2: Replace ‘quicker’ with ‘faster’.

Figure 1 is not of the same high quality as the other figures, I think the data can be shown in a way that is more visually appealing and informative way.

References:

Sobolev, S.V., Sobolev, A.V., Kuzmin, D.V., Krivolutsкая, N.A., Petrunin, A.G., Arndt, N.T., Radko, V.A., Vasiliev, Y.R., 2011. Linking mantle plumes, large igneous provinces and environmental catastrophes. *Nature* 477, 312–316.

Jones, M.T., Jerram, D.A., Svensen, H.H., Grove, C., 2016. The effects of large igneous provinces on the global carbon and sulphur cycles. *Palaeogeography, Palaeoclimatology, Palaeoecology* 441(1), 4-21.

Reviewer #3 (Remarks to the Author)

I have completed my review of manuscript NCOMMS-16-25873 entitled “Initial Siberian Traps sill emplacement as the trigger of the end-Permian mass extinction”, submitted by Burgess et al. to *Nature Communications*.

Several of the Phanerozoic “Big Five” mass extinctions have been linked to simultaneous large-volume magmatism from so-called Large Igneous Provinces, or LIPs. In recent years, the case for a temporal link has grown stronger, as geochronological techniques have improved; and the causal link has been strengthened as well, as our understanding of the mechanisms of extinction has expanded. Perhaps the least controversial of these LIP-extinction pairs is the end-Permian extinction and the Siberian Traps, which are the focus of the manuscript.

The principal claim of the authors rests on the observation that, in spite of the close temporal link between LIPs and extinctions, the time scale of extinction (on the order of a few tens of thousands of years) remains widely different from the timing of eruption (typically on the order of a few hundred thousands years at best, and well over a million years in some cases). Thus, only a fraction of the magmatic event can be implicated in the brief interval of extinction. Based on high precision geochronology, the authors argue that a rapid transition in LIP emplacement from surface flows to crustal sills is most closely associated in time with the extinction, and that the sudden release of greenhouse gases from sediments during these sill intrusions would have provided the trigger for the extinction.

This topic is fascinating and of global interest to the scientific community at large; in addition, it is timely, and fits nicely in current renewed conversations about the role of LIPs in mass extinctions. I really wanted to offer this manuscript a sterling review; however, in my opinion, it falls short on two accounts.

The most problematic aspect is that, as far as I can tell, it does not present any new data. The data presented in Fig. 2 are from Burgess & Bowring (*Science Advances*, 2015). Nearly all of the “results”, particularly the first two paragraphs of the section, are not “results” per se, but a summary of Burgess & Bowring (2015). More problematically, the idea that the mass extinction was coincident with the first sill emplacement is discussed explicitly by Burgess & Bowring (2015), and I quote: “The mass extinction interval ends at 251.880 ± 0.031 Ma, slightly postdating, but within uncertainty of emplacement of the oldest dated sample of the Noril’sk 1 sill, the oldest sill dated from anywhere in the province”. Later on in the same paper, Burgess & Bowring (2015) consider plausible the idea that sill intrusion is the main culprit in the release of greenhouse gases and corresponding environmental changes (“because there is evidence for extrusive magmatism

~300 ka before the onset of mass extinction, it is probable that this extrusive magmatism had an intrusive counterpart now covered by younger lavas and pyroclastic rocks. Thus, the model of [Svensen et al. 2009] is plausible and preferred here"). Clearly, both the idea of a close temporal relationship between sill emplacement and mass extinction, and the data to support it, already exist in the literature.

Another problematic aspect is the idea that this mechanism could be applied to other LIP-extinction pairs. In particular, this mechanism is clearly not applicable to the Deccan, where very few sills have been documented, and the substrate is granitic without potential for release of substantial greenhouse gases (granted, the Chicxulub impact substantially complicates the argument).

In conclusion, the authors' claims are convincing and the manuscript is well written, but their argument is not novel and was already made, though not as explicitly, in their Burgess & Bowring 2015 paper.

Other comments:

- "GSSP" at the top of page 3 is an undefined acronym.
- Fig. 1 and its caption are incomplete. "EPE", "ETE", and "ECE" are not defined. The difference in color between the bars is not explained in the caption. It took me a couple of minutes to decipher "ErJE OAE" as "Early Jurassic Ocean Anoxic Event".

Review by Richard Ernst of Burgess, Muirhead and Bowring "Initial Siberian Traps sill emplacement as the trigger of the end-Permian mass extinction"

GENERAL COMMENTS

This is another breakthrough publication by Seth Burgess and colleagues. I heard (and was impressed by) their talk at the annual GSA meeting (Sept 2016) on the theme that has been developed into this full paper. Glad to see these new insights developed into a full paper. Previously the high precision U-Pb dating by Burgess and Bowring (Science Advances) on about 25 samples from the Siberian trap event demonstrated incredibly short duration of the LIP event and the overlap with the mass extinction. At the time it was stunning that the Siberian Trap event was shown to have a duration of less than 1 myr. With this current manuscript (submitted for Nature Communications) Seth Burgess has taken the data interpretation to a new level of being able to link the extinction to a specific point in the overall (short) 1 myr duration. The notion that the extinction was linked to the start of the intrusive-only phase of the Siberian trap LIP is consistent with the emerging evidence from the Oslo group (e.g Svensen et al. 2009) for the importance (even dominance) in many LIPs of gas release from interaction of sills with volatile-rich host rocks and transport of these gases to the atmosphere through Hydrothermal Vent Complexes.

DETAILED COMMENTS

It is too bad that there are NOT line numbers as this would facilitate more easy commentary.

Line numbers now added.

ABSTRACT: "However, LIP magmatism is long-lived relative to extinction.." In this new 'universe' that Burgess and other high precision U-Pb daters are bringing us to— a less than one myr duration for a LIP event is becoming normal (rather than shockingly short). However, there are also some LIP events that in fact are much longer and consist of multiple short pulses that potentially span 10s of myr-- e.g. 1115-1085 Ma Keweenawan LIP of North America (Ernst 2014).

We agree that there are LIPs characterized by much longer emplacement durations, and that the Keweenawan qualifies as one. We have modified the abstract (L 18) to reflect the prolonged nature of some LIPs.

INTRODUCTION:

Could I suggest referencing the only text book on LIPs (Ernst, R.E. 2014. Large Igneous Provinces. Cambridge University Press, 653 p.)—and perhaps adding this citation just after LIPs are first mentioned: "Large igneous province (LIP) magmatism" (Ernst, 2014)

This reference has been added (L 33).

“The first is a significant disparity in the timescales over which LIP magmatism and mass extinctions occur” It is again so noteworthy that Burgess (and other geochronologists) have shown us that a 1 million year duration LIP event is in fact LONG compared to the duration of a mass extinction. What an exciting new world of LIP geochronology we are in now!

We fully agree, and believe this paper is an excellent opportunity to utilize this high-resolution data to pinpoint the causal association between magmatism and mass extinction.

In the Introduction: “Thirdly, not all LIP events are associated with marked environmental change” and the reference used here is Stothers 1993. Can a more recent reference also be added: e.g. Bond, David P.G., Grasby, Stephen E., On the causes of mass extinctions, Palaeogeography, Palaeoclimatology, Palaeoecology (2016), doi: 10.1016/j.palaeo.2016.11.005 Or Courtillot, V.E., Renne, P.R., 2003. On the ages of flood basalt events. Comptes Rendus Geosciences 335, 113–140 [I have a paper explicitly on this theme of linking of LIP events and environmental impacts for PPP, but unfortunately it is a few weeks away from being accepted.]

The Bond and Grasby reference has been added (L 59).

DISCUSSION section, third paragraph:

You are proposing that Stage 1 lavas represent a “structural barrier that inhibit further magma ascent to the surface”. This would imply that the first sills should be emplaced right below the flood basalt package and that the sill complex should progressively build downward (i.e. deeper and deeper in the crust). This seems to be implied in Figure 3.

We have removed the term “structural barrier” from the paper. It seems this term has given the impression that the load itself creates a physical barrier, which the initial sill intrudes below. This sill then impedes the ascent of each successive intrusion, thus produces a downward growth model for the sill complex.

This is not the model invoked under a loading scenario (e.g., Pinel and Jaupart, 2000). Instead, the load creates a compressive stress state in the upper few kilometers of the crust, which has the potential to inhibit the ascent of dikes. Under a crustal loading model (e.g., Pinel and Jaupart, 2000), the exact depth at which magma ascent will be impeded will vary through time. It will depend on both the evolving properties of the load (as function of density and thickness) and changing driving pressures of the magma, controlled by overpressure at the source and magma buoyancy (product of both magma and country rock density). The depth where magma is “trapped”, and then spreads laterally, will therefore vary vertically through time. However, we have no reason to assume that it will

move progressively downward as suggested by the reviewer.

An interesting question is what thickness the flood basalt package needs to attain before the flood basalt package begins to act a “structural barrier”

This is an interesting problem that has been explored previously in Pinel and Jaupart (2000). Their work shows that increasing load thickness will increase the chances of magma stalling at depth. However, the thickness at which a volcanic load will inhibit magma ascent will depend strongly on magma driving pressures (magma buoyancy + source overpressure), and hence a lava load of a few hundred meters can be equally as capable of trapping dikes as a two kilometer volcanic pile, depending on the magma properties. Any discussion on this point is beyond the scope of the current study, and we instead refer the reader to work of Pinel and Jaupart (2000) (L 154) who provide solutions to this problem for a number of scenarios.

“crustal loading / unloading” model: I am not understanding this model. I can see the potential impact of erosion to cause unloading, leading to sill emplacement. Is this what you have in mind? Or are you producing the sills during the crustal loading stage—I don’t see how

“Unloading” has been deleted from the text, as this term referred to stress rotations from faulting during continental rifting. Our model instead invokes crustal loading to shift from dominantly sub-vertical (dikes) to sub-horizontal (sills).

Does the chemistry of Stage 1 magmatism differ from Stage 2 magmatism? I would think that the initial lavas which ascend directly to the surface are less likely to exhibit any crustal chemical contamination. However, the Stage 2 intrusives will have more significant interactions with their host rocks (to produce the gases associated with extinction) and should potentially be more crustally contaminated. Is there any geochemistry data that compares the flood basalts of Stage 1 with the intrusives of Stage 2. In this regard it is interesting that the Nadezhdinsky lavas (part of Stage 1) do exhibit crustal contamination. So it would seem that at least some Stage 1 lavas still did pause and interact with continental crust, before ascending to be emplaced as flows.

In short, there is no reason to assume a correlation exists between the measurable degree of crustal contamination (via geochemistry) and the rate at which LIP magma ascends the crust en route to eruption or sill formation. The assertion is supported by the fact that stage 1 and 2 magmas are not geochemically distinct, on average. Stage 2 sills are remarkably invariant in chemistry, and are similar to the majority of stage 1 lavas. There are variations in lava chemistry in stage 1, for example the silica-undersaturated melenephenelites; however, these non-tholeiitic basalts are volumetrically very minor. The main thrust of this comment is the question of whether significant

assimilation of crustal material is evident in sill geochemistry, and if so, is it markedly different than that seen in lavas. Again, the short answer to this question is no. Firstly, sills are not so large as to metasomatise and assimilate large volumes of country rock. They do impart heat and generate a contact aureole (the place from which volatiles are generated), but the bulk chemistry of the sill is relatively unchanged in this process but for a thin cooling rind on the exterior of the sill. Secondly, the relatively rapid crustal transit of lavas does not preclude some crustal contamination, but caution must be exercised when using a basalt from the Noril'sk section (as this comment does), as evidence of such contamination. This region is characterized by significant syn/post magmatic mineralization, a trait not representative of other basalts throughout the province, which show very little crustal contamination, if any.

The comment by the Dr. Ernst alludes to an interesting idea; that sill chemistry relative to lava chemistry could tell us the degree to which LIP rock interacted with the host rock, and thus be used as a test of the model presented here. Unfortunately, sill geochemistry is relatively unaffected by the shallow-crustal contact metamorphic processes necessary to generate volatiles, and is not readily differentiable from the vast majority of lava geochemistry.

At the end of this paragraph you note “the striking outcrop pattern... which is characterized by extrusive rocks in the interior of the province and a zone of younger sills flanking these rocks around nearly the entire periphery of the province” You consider this to be a primary pattern, but I wonder if this pattern could simply represent deeper erosion (and removal of the flood basalts) at the periphery of the province.

We have deleted this observation from the text, as it is difficult to precisely constrain the degree to which erosion has affected the province since deposition. That said, although one would think significant erosion has occurred in the LIP over the past ~250 Ma, there is actually evidence for little incision into the province. Erosion is confined mainly to river valleys. The magnitude of erosion that characterizes the Anabar region, as mentioned by the reviewer below, is not common. Basement is only exposed at the margins of the craton, and in two shield areas, one of which is the Anabar, but these areas are an exception. The LIP erupted onto stable cratonic lithosphere, which shows very little evidence for structural deformation and uplift, and related erosion.

Importantly, even if erosion did shed significant basalt volume from atop the sills, our model is unaffected. The relative timing of sill and lava emplacement still remains, as does the 3-stage emplacement framework, hiatus in eruption, relative timing of magmatism and mass extinction, and thus, the driving force of our model.

DISCUSSION section, fourth paragraph:

In your discussion of the location of the feeder system to the Stage 1 lavas, you

could note that we have identified a giant radiating and giant circumferential swarm that both locate a mantle plume centre just to the north of the main volcanics, at a location of approximately 100 E, 72 N (Ernst and Buchan, 2001 GSA Special Paper 352, p. 247-265; see also Figure 16.4 in Ernst, 2014)

This suggestion and reference have been added (L 168).

DISCUSSION section

An additional consideration is that if the accumulation of Stage 1 lavas represents a structural barrier that leads to a stage 2 dominated by intrusions, then what changes to allow lava flows to be again emplaced at the following stage 3?

As requested, we speculate briefly on what might have allowed for renewed eruptions (L161-163).

FIGURE CAPTIONS

Figure 1 Caption: in the caption you need add a link to the colours, blue and red. So after “mass extinction” add “(blue)” and after “large igneous province (LIP)” add “red”. In the caption explain: EPE, ETE, ErJE, OAE and ECE. The key reference for the U-Pb dating for the Karoo portion of the Karoo-Ferrar LIP is missing: Add this reference: Svensen, H., Corfu, F., Polteau, S., Hammer, Ø., Planke, S., 2012. Rapid magma emplacement in the Karoo Large Igneous Province. Earth Planet. Sci. Lett. 325–326, 1–9. doi 10.1016/j.epsl.2012.01.015.

These suggestions have been incorporated into the figure caption, and the additional reference has been added (L 377).

Figure 2 caption: explain the stars in the carbon isotope record

The figure caption has been amended and a reference has been added (L 387).

FIGURE 1: Very powerful diagram

In response to another reviewer, we have significantly revised this figure, although the core information the figure attempts to convey is unchanged.

FIGURE 3: this model assumes no erosion of flood basalts from the periphery of the province (where the sills are dominant—see discussion above). However, we know that major erosion occurred in the vicinity of the Anabar shield. A mantle plume centre could be added to the diagram (based on Ernst and Buchan 2001 or in Ernst 2014) at a location of approximately 100 E, 72 N

We are familiar with Dr Ernst's hypothesis of a plume head in the Anabar area. Advocating for this particular hypothesis would detract from the main point of the

figure, which is to present a conceptual cross-section of lavas eruption, sill deflection, and plumbing system development during stages 1-3, and related greenhouse gas generation. The existence, or non-existence, of a plume in the Anabar has no bearing on our model.

Reviewer #2 (Remarks to the Author):

Review of Burgess et al: Initial Siberian Traps sill emplacement as the trigger of the end-Permian mass extinction

This manuscript offers a new and insightful hypothesis for the volcanogenic mechanism that drove the end-Permian extinction. The hypothesis is novel, important for the understanding of the relationship between large igneous provinces and mass extinction events, anchored firmly in the state-of-the-art dataset now available, and is therefore worthy of consideration for this journal. I have a few issues with the manuscript in its current form, which may affect the final publication of the manuscript.

Major Points:

1) While the observation that the first pulse of sill emplacements into the Tunguska basin is coincident with the end-Permian mass extinction is new and extremely important, this paper does not currently have any 'new' results. The current Results Section is a description of the work the authors and others have done previously. I don't think this negates this from being published in Nature Communications, but I do feel that it needs to be restructured somewhat to reflect that this is a new hypothesis to explain existing data.

We have changed this section name to "relative timing," which we think better describes its function. We have also addressed this comment in greater detail in our general response to reviewers, as discussed in "Response to specific Editor comments."

2) The argument put forward is based on the remnant outcrops of the Siberian Traps in the Tunguska Basin. There needs to be a few sentences added to address the certainty that the authors feel they have that the whole story is covered by the surviving exposures.

As stated in response to Reviewer 1, we have deleted reference to the map pattern of the province.

3) Discussion: This section makes the assumption that LIP lavas are too low in CO₂ to be a primary source of carbon. Calculated CO₂ volumes of individual lavas may not be representative of total magmatic degassing from LIPs. For example, the arrival of a plume head may be considerably richer in volatiles compared to lavas that came later in the LIP emplacement (this doesn't fit the

data presented in this paper, but needs to be discussed nonetheless). Moreover, the thermal alteration of the lithosphere as the plume head arrived beneath the crust may have led to considerable degassing of CO₂ (e.g. Sobolev et al, Nature, 2011). Can the authors be certain that this slightly delayed pulsed isn't due to this volatile rich front reaching the surface? I understand that this paper is proposing an alternative hypothesis, but the text needs to be changed to reflect that this is one of several possible avenues that could explain the carbon isotope excursion.

We have added content to section 1, “the case for causation” (L 47-48) and section 2 “identifying the smoking gun” (L 122-128) to include the Sobolev et al., (2011) model, and have briefly discussed why this model is incompatible with the temporal relationship between magmatism and mass extinction. In short, the revised geochronology of Burgess et al. (2014) reveals that plume head arrival from Sobolev et al. (2011) (and volatile front arrival at the surfaces) precedes mass extinction by ~900 Ka. The timing just doesn't work.

Minor Points:

Abstract: 'Sub-interval' instead of 'sub interval'.

This comment has been addressed in the text

Introduction, Paragraph 1: References 1 and 2 are out of order

The reference list has been amended to address this comment.

Introduction, Paragraph 1: The greenhouse gas H₂O is released in negligible quantities compared to surface reservoirs, and will not affect the radiative properties of the Earth as a primary driver, only as a positive feedback to other greenhouse gas emissions (see the review of Jones et al, 2016 and references therein).

Water has been taken out of the list of greenhouse gasses, and the Jones et al. reference has been added (L 47).

Introduction, Paragraph 2: There is an implicit assumption in this writing that the mass extinction can only be caused by volcanism. It is highly likely that it was, but I think a degree of caution with the language used here would be prudent.

We have amended the text at the beginning of section 1 “the case for causation” to reflect other plausible trigger mechanisms (namely bolide impact), and have added a reference supporting this mechanism (Becket et al) (L 38). We would also like to point to the concluding thoughts of this section (L 72), wherein we suggest that the possibility exists that no part of LIP magmatism is responsible for triggering extinction. Our language here is not meant to be over-confident, but to reflect the lack of tenable alternate trigger mechanisms.

Discussion, Paragraph 1: End sentence, recommend changing to ‘...qualifying aliquot is dominated by intrusive sources.’

In response to other comments, this section has been condensed. The line to which the reviewer refers in this instance has been cut.

Discussion, Paragraph 2: Replace ‘quicker’ with ‘faster’.

In response to other comments, this section has been condensed. The line to which the reviewer refers in this instance has been cut.

Figure 1 is not of the same high quality as the other figures, I think the data can be shown in a way that is more visually appealing and informative way.

In an effort to make this figure more visually appealing, we have changed the orientation of the figure and added additional text for clarity.

References:

Sobolev, S.V., Sobolev, A.V., Kuzmin, D.V., Krivolutskaya, N.A., Petrunin, A.G., Arndt, N.T., Radko, V.A., Vasiliev, Y.R., 2011. Linking mantle plumes, large igneous provinces and environmental catastrophes. Nature 477, 312–316.

Jones, M.T., Jerram, D.A., Svensen, H.H., Grove, C., 2016. The effects of large igneous provinces on the global carbon and sulphur cycles. Palaeogeography, Palaeoclimatology, Palaeoecology 441(1), 4-21.

These references have both been added, as discussed above.

Reviewer #3 (Remarks to the Author):

I have completed my review of manuscript NCOMMS-16-25873 entitled “Initial Siberian Traps sill emplacement as the trigger of the end-Permian mass extinction”, submitted by Burgess et al. to Nature Communications.

Several of the Phanerozoic “Big Five” mass extinctions have been linked to simultaneous large-volume magmatism from so-called Large Igneous Provinces, or LIPs. In recent years, the case for a temporal link has grown stronger, as geochronological techniques have improved; and the causal link has been strengthened as well, as our understanding of the mechanisms of extinction has expanded. Perhaps the least controversial of these LIP-extinction pairs is the end-Permian extinction and the Siberian Traps, which are the focus of the manuscript.

The principal claim of the authors rests on the observation that, in spite of the close temporal link between LIPs and extinctions, the time scale of extinction (on the order of a few tens of thousands of years) remains widely different from the timing of eruption (typically on the order of a few hundred thousands years at best, and well over a million years in some cases). Thus, only a fraction of the magmatic event can be implicated in the brief interval of extinction. Based on high precision geochronology, the authors argue that a rapid transition in LIP emplacement from surface flows to crustal sills is most closely associated in time with the extinction, and that the sudden release of greenhouse gases from sediments during these sill intrusions would have provided the trigger for the extinction.

This topic is fascinating and of global interest to the scientific community at large; in addition, it is timely, and fits nicely in current renewed conversations about the role of LIPs in mass extinctions. I really wanted to offer this manuscript a sterling review; however, in my opinion, it falls short on two accounts.

The most problematic aspect is that, as far as I can tell, it does not present any new data. The data presented in Fig. 2 are from Burgess & Bowring (Science Advances, 2015). Nearly all of the “results”, particularly the first two paragraphs of the section, are not “results” per say, but a summary of Burgess & Bowring (2015). More problematically, the idea that the mass extinction was coincident with the first sill emplacement is discussed explicitly by Burgess & Bowring (2015), and I quote: “The mass extinction interval ends at 251.880 ± 0.031 Ma, slightly postdating, but within uncertainty of emplacement of the oldest dated sample of the Noril’sk 1 sill, the oldest sill dated from anywhere in the province”. Later on in the same paper, Burgess & Bowring (2015) consider plausible the idea that sill intrusion is the main culprit in the release of greenhouse gases and corresponding environmental changes (“because there is evidence for extrusive magmatism ~300 ka before the onset of mass extinction, it is probable that this extrusive magmatism had an intrusive counterpart now covered by younger lavas and pyroclastic rocks. Thus, the model of [Svensen et al. 2009] is plausible and preferred here”). Clearly, both the idea of a close temporal relationship between sill emplacement and mass extinction, and the data to support it, already exist in the literature.

The concern over novelty is warranted here, and we regret not being clearer in the manuscript about the large step forward this new idea represents, and how it builds on previously published work. The “results” of this manuscript are not new geochronology, but rather a robust model by which, for the first time, temporally associated LIPs and mass extinctions are united with the widely accepted mechanism by which LIPs trigger extinction (volatile release via contact metamorphism).

As this reviewer correctly points out, a synopsis of the 2015 Science Advances datasets is included in the newly-named “relative timing” section. We defend this

inclusion, as this synopsis builds the temporal framework for our new model. Without this context, we feel as though readers will be left requiring far too much prerequisite (and detailed) knowledge of the relative timing of magmatism and mass extinction. We have trimmed this section considerably by combining what were two paragraphs into one (L 99-112). We believe this greatly improves the flow and clarity of this section, while retaining all the original and critical scientific content.

In the 2015 *Science Advances* paper, geochronology on the Siberian LIP was presented, which outlined an age model for both mass extinction and LIP magmatism, and suggested broad temporal coincidence between the two. In the intervening years, many studies have used this broad temporal coincidence to assert causation – however, as we point out with this current submission, simple coincidence does *not* guarantee causation, and a void exists between the two. We believe our model bridges this void, as it is the first to explain which part of LIP magmatism drove collapse, and more importantly, why.

The revised text illustrating this point, and addressing this comment, is highlighted in the “Response to specific Editor comments” section above.

Another problematic aspect is the idea that this mechanism could be applied to other LIP-extinction pairs. In particular, this mechanism is clearly not applicable to the Deccan, where very few sills have been documented, and the substrate is granitic without potential for release of substantial greenhouse gases (granted, the Chicxulub impact substantially complicates the argument).

This is an astute comment. We believe the efficacy of a LIP in driving extinction is directly tied to the proportion of sills relative to lavas, a hypothesis we have added to the manuscript in a short section, “making a deadly LIP,” at the end of the paper (L 190 -209).

In conclusion, the authors’ claims are convincing and the manuscript is well written, but their argument is not novel and was already made, though not as explicitly, in their Burgess & Bowring 2015 paper.

As discussed in the “Response to specific Editor comments” section above and the reviewer comments, the model we present, which is built on the Burgess and Bowring (2015) dataset, is wholly new. This work highlights a unique geochronology dataset in terms of the quality and resolution, generated using identical procedures and laboratory equipment, permitting comparison at the level of analytical uncertainty. By identifying a robust temporal coincidence between initial sill-complex emplacement and mass extinction, we pinpoint for the first time the deadly magma aliquant for a LIP. This submission also indicates for the first time that LIPs characterized by sill-complexes are more likely to trigger catastrophic global environmental change than their flood basalt- and/or dike-dominated counterparts.

Other comments:

- *“GSSP” at the top of page 3 is an undefined acronym.*
- *Fig. 1 and its caption are incomplete. “EPE”, “ETE”, and “ECE” are not defined. The difference in color between the bars is not explained in the caption. It took me a couple of minutes to decypher “ErJE OAE” as “Early Jurassic Ocean Anoxic Event”.*

The caption for figure 1 has been amended to address this comment, and GSSP has been defined in the text (L 100).

Reviewers' Comments:

Reviewer #1:

Remarks to the Author:

Comments by Richard Ernst on the revised version of Burgess et al "Initial Siberian Traps sill emplacement as the trigger of the end-Permian mass extinction"

COMMENTS

This paper is ready to be accepted, after consideration of the minor suggestions in this re-review.

The additional changes made in response to the comments of myself and the other two reviewers are welcome and give the paper more depth and impact.

For instance, the novel aspects of the model are now more emphasized and the details of the dating story decreased, so that the paper more correctly builds on (and does not repeat details of) the content already presented in the previous geochron paper in Science Advances.

The ending sentence succinctly captures the huge significance of this paper:

"The model presented here suggests that LIPs characterized by widespread initial sill emplacement into a volatile-fertile basin are lethal on a global scale"

I think the word "initial" could be dropped since in this Siberian example the flood basalts represent the initial pulse not the sills.

Here is a suggested rewrite of that sentence: "The model presented here suggests that LIPs characterized by A widespread PULSE OF sill emplacement into a volatile-fertile basin are lethal on a global scale"

Regarding the idea that the developing of a volcanic load (Stage 1) leads to suppression of upward dyke transport and enhanced sill production (Stage 2):

This idea is linked to the Pinel and Jaupart (2000) paper. I have looked up this paper and have started to work through it. The modeling in P and J (2000) suggests that dyke ascent is suppressed whether the volcanic load is conical (like a shield volcano) or slab-like (more like flood basalts).

My question would be whether the areal extent of the slab matters. If the slab (i.e. flood basalt pile) is widespread—over many hundreds of km, then it would seem to me that in the centre of this slab (far from its edges) that it would simply represent just a thicker crust, and dyke ascent would be the same as where the crust was thinner. Only towards the edges of the slab (of the flood basalt pile) would the difference in thickness with the adjacent crust matter, leading to dyke suppression. So perhaps the distribution of the flood basalts would be critically important to the location of areas of suppressed dyke ascent. So in areas of widespread continuous flood basalts the suppression would only occur at the edge of the slab-load (at the edge of the flood basalt pile). Or if the flood basalts are more discontinuously distributed (with local areas of no flood basalts) then the dyke suppression could be more widespread and would be concentrated adjacent to these areas of absent flood basalts.

Lines 18-19, Abstract: "LIPs are long-lived compared to mass extinctions (1-50 Ma vs. <100 ka)."
In the most general sense this is true, but single pulsed LIPs are always very short (probably 1-5 myr) and multi-pulsed LIPs only reach 50 myr in the rarest of cases. I would suggest rewriting. Perhaps replace part of this sentence with "Single pulsed LIPs are 1-5 myr and multi-pulsed LIPs can span several 10s of myr."

See also lines 51-53 for the same point.

Figure 2 remains such a compelling picture of how the 3 stages correlate with the pattern in the

carbon isotopes. Really clear and impressive story.

I am intrigued by the additional significant negative carbon excursion in the middle of Stage 3 (at 251.4 Ma).

In my original review I commented on the location of the plume centre, in your reply it seems that my point was not clear. Actually, I was NOT implying that the plume centre was located under the Anabar shield uplift. Rather the plume centre is located to the N of the main distribution of the flood basalts in the Yenesei-Khatanga trough at the focus of a radiating dyke swarm. I am currently unsure of the reason for the Anabar shield uplift (which was presumably originally overlain by a continuation of the flood basalt pile and now exposes part of the regional radiating dyke swarm that fed the flood basalts).

In my original review I had mentioned a new review paper that was not yet accepted; it is now in press and could be cited alongside the Kidder and Worsley (2010) or the Bond and Grasby (2016) paper.

Ernst, R.E. and Youbi, N., 2017, How Large Igneous Provinces affect global climate, sometimes cause mass extinctions, and represent natural markers in the geological record. *Palaeogeography, Palaeoclimatology, Palaeoecology*, (in press), doi: 10.1016/j.palaeo.2017.03.014

This Burgess et al. manuscript represents an important scientific contribution and I look forward to it being in press so that I can start citing it.

Reviewer #2:

Remarks to the Author:

The revised submission from Burgess and colleagues is a much improved manuscript, both in layout and content. The responses to reviewers are thorough, fair, and the arguments well presented. The manuscript text reads well, the section headings are more appropriate for this type of "new hypothesis" paper, and I believe that it is now worthy of publication in *Nature Communications*.

I only have one minor comment with regards to figure 1. For the four LIPs mentioned, there is evidence for volcanism/magmatism before the extinction events in each case. It may not fit with how the authors wish to display the data, but if it is just a comparison of durations it is misleading to have the duration of extinction at the beginning of each LIP duration. It would also be improved by adding the error bars to both the LIP and Extinction durations.

Apart from this, I believe this to be an excellent piece of work.

With best regards

Morgan Jones

Reviewer #3:

Remarks to the Author:

In my original review of the manuscript entitled "Initial Siberian Traps sill emplacement as the trigger of the end-Permian mass extinction" by Burgess and colleagues, I raised concerns related to the novelty of the argument presented, and of the data to support it. As this fairly harsh assessment was a dissenting view compared to that of other reviewers, it could have been easily dismissed by the authors.

I am thrilled to see the authors took my concerns seriously in their revisions of the manuscript.

The novelty of their model is highlighted much more clearly in the revised version of their ms., and they have made significant improvements both in terms of the paper's structure, and formulation of their argument. With the exception of minor typos and editing errors (for instance, the way references are cited on line 79 and elsewhere, in parentheses instead of superscripts; or the absence of a closing parenthesis at the end of line 128), I would recommend the manuscript be accepted without further revisions.

Author responses are shown in blue:

Reviewer #1 (Remarks to the Author):

Comments by Richard Ernst on the revised version of Burgess et al “Initial Siberian Traps sill emplacement as the trigger of the end-Permian mass extinction”

COMMENTS

This paper is ready to be accepted, after consideration of the minor suggestions in this re-review.

The additional changes made in response to the comments of myself and the other two reviewers are welcome and give the paper more depth and impact.

For instance, the novel aspects of the model are now more emphasized and the details of the dating story decreased, so that the paper more correctly builds on (and does not repeat details of) the content already presented in the previous geochron paper in Science Advances.

The ending sentence succinctly captures the huge significance of this paper:
“The model presented here suggests that LIPs characterized by widespread initial sill emplacement into a volatile-fertile basin are lethal on a global scale”
I think the word “initial” could be dropped since in this Siberian example the flood basalts represent the initial pulse not the sills.
Here is a suggested rewrite of that sentence: “The model presented here suggests that LIPs characterized by A widespread PULSE OF sill emplacement into a volatile-fertile basin are lethal on a global scale”

This is an excellent point. The suggested wording for this final sentence has been added to the manuscript.

Regarding the idea that the developing of a volcanic load (Stage 1) leads to suppression of upward dyke transport and enhanced sill production (Stage 2):
This idea is linked to the Pinel and Jaupart (2000) paper. I have looked up this paper and have started to work through it. The modeling in P and J (2000) suggests that dyke ascent is suppressed whether the volcanic load is conical (like a shield volcano) or slab-like (more like flood basalts). My question would be whether the areal extent of the slab matters. If the slab (i.e. flood basalt pile) is widespread—over many hundreds of km, then it would seem to me that in the centre of this slab (far from its edges) that it would simply represent just a thicker crust, and dyke ascent would be the same as where the crust was thinner. Only towards the edges of the slab (of the flood basalt pile) would the difference in thickness with the adjacent crust matter, leading to dyke suppression.

We do not agree with the assertion that a widespread volcanic load with little topography would only affect the stress state at the edges of the field, which would contrast modelling studies investigating this issue. For example, numerical modeling by Le Corvec et al. (2015) shows that volcanic loads that have radii on the order of 100s of kilometers with slopes of only 1 to 2 degrees effect stress states below the entire load.

Le Corvec, N., McGovern, P. J., Grosfils, E. B., & Galgana, G. Effects of crustal-scale mechanical layering on magma chamber failure and magma propagation within the Venusian lithosphere. *Journal of Geophysical Research: Planets*, **120(7)**, 1279-1297 (2015).

So perhaps the distribution of the flood basalts would be critically important to the location of areas of suppressed dyke ascent. So in areas of widespread continuous flood basalts the suppression would only occur at the edge of the slab-load (at the edge of the flood basalt pile). Or if the flood basalts are more discontinuously distributed (with local areas of no flood basalts) then the dyke suppression could be more widespread and would be concentrated adjacent to these areas of absent flood basalts.

These are interesting points, however we cannot constrain the thickness distribution of the flood-basalt prior to erosion. Any discussion along these lines would be speculative and beyond the scope of the paper, which focuses on the geochronology of different magmatic events and their connection with environmental changes and mass extinction.

Finally, we would like to point out that the reviewer is touching on some key unresolved issues regarding the occurrence of sill complexes in both the Siberian Traps LIP, and also LIPs generally. The causes for sill formation are frequently debated (e.g., Thomson and Hutton, 2004; Kavanagh et al., 2005; Magee et al., 2016; Walker et al., 2017), and hence *we highlight 3 possible mechanisms* to alter the state of stress to drive sill formation in the Siberian LIP (loading, high magma flux, and mechanical layering effects). Our uncertainty regarding this issue is acknowledged in the original text on lines 148-149, where we explicitly state that the exact cause for the observed transition to widespread sill complex formation is challenging to pinpoint.

Finally, we do not claim that sill complex formation was definitely the result of volcanic loading. We only state that sill formation occurred after a volcanic load was established, and that this highlights the possibility that this loading drove sill formation, as this is an accepted mechanism. However, we then describe other mechanisms through which sill formation may have occurred with relevant referencing.

For these reasons, we do not favor modifying the text to accommodate these comments.

Walker, R. J., Healy, D., Kawanzaruwa, T. M., Wright, K. A., England, R. W., McCaffrey, K. J. W., & Blenkinsop, T. G. Igneous sills as a record of horizontal shortening: The San Rafael subvolcanic field, Utah. *Geological Society of America Bulletin*, **B31671-1** (2017).

Kavanagh, J. L., Menand, T., & Sparks, R. S. J. An experimental investigation of sill formation and propagation in layered elastic media. *Earth and Planetary Science Letters*, **245(3)**, 799-813 (2006).

Thomson, K., & Hutton, D. Geometry and growth of sill complexes: insights using 3D seismic from the North Rockall Trough. *Bulletin of Volcanology*, **66**(4), 364-375 (2004).

Magee, C., Muirhead, J. D., Karvelas, A., Holford, S. P., Jackson, C. A., Bastow, I. D., & Shtukert, O. Lateral magma flow in mafic sill complexes. *Geosphere*, **12**(3), 809-841 (2016).

Lines 18-19, Abstract: "LIPs are long-lived compared to mass extinctions (1-50 Ma vs. <100 ka)." In the most general sense this is true, but single pulsed LIPs are always very short (probably 1-5 myr) and multi-pulsed LIPs only reach 50 myr in the rarest of cases. I would suggest rewriting. Perhaps replace part of this sentence with "Single pulsed LIPs are 1-5 myr and multi-pulsed LIPs can span several 10s of myr."

See also lines 51-53 for the same point.

We have deleted this duration range from the abstract in an effort to shorten the text, and have added text in the body of the manuscript (line 99) to address this comment.

Figure 2 remains such a compelling picture of how the 3 stages correlate with the pattern in the carbon isotopes. Really clear and impressive story. I am intrigued by the additional significant negative carbon excursion in the middle of Stage 3 (at 251.4 Ma).

This excursion is indeed compelling, but not the specific focus of this manuscript.

In my original review I commented on the location of the plume centre, in your reply it seems that my point was not clear. Actually, I was NOT implying that the plume centre was located under the Anabar shield uplift. Rather the plume centre is located to the N of the main distribution of the flood basalts in the Yenesei-Khatanga trough at the focus of a radiating dyke swarm. I am currently unsure of the reason for the Anabar shield uplift (which was presumably originally overlain by a continuation of the flood basalt pile and now exposes part of the regional radiating dyke swarm that fed the flood basalts).

We appreciate the clarification.

In my original review I had mentioned a new review paper that was not yet accepted; it is now in press and could be cited alongside the Kidder and Worsley (2010) or the Bond and Grasby (2016) paper.

Ernst, R.E. and Youbi, N., 2017, How Large Igneous Provinces affect global climate, sometimes cause mass extinctions, and represent natural markers in the geological record. *Palaeogeography, Palaeoclimatology, Palaeoecology*, (in press), doi: 10.1016/j.palaeo.2017.03.014

This reference (now #3) has been added to the manuscript.

This Burgess et al. manuscript represents an important scientific contribution and I look

forward to it being in press so that I can start citing it.

Reviewer #2 (Remarks to the Author):

The revised submission from Burgess and colleagues is a much improved manuscript, both in layout and content. The responses to reviewers are thorough, fair, and the arguments well presented. The manuscript text reads well, the section headings are more appropriate for this type of "new hypothesis" paper, and I believe that it is now worthy of publication in Nature Communications.

I only have one minor comment with regards to figure 1. For the four LIPs mentioned, there is evidence for volcanism/magmatism before the extinction events in each case. It may not fit with how the authors wish to display the data, but if it is just a comparison of durations it is misleading to have the duration of extinction at the beginning of each LIP duration. It would also be improved by adding the error bars to both the LIP and Extinction durations.

We agree with this observation of the relative timings of magmatism and extinction, but stand by the construction of the figure, which is meant specifically to highlight the duration disparity between the two events. To help clarify, we have amended the figure caption to include this reviewer point. Addition of error bars to each duration is inappropriate in this case, as these are (in most cases) estimates informed by data from multiple studies rather than a single dataset with an associated uncertainty. We have amended the figure caption to clarify this point.

Apart from this, I believe this to be an excellent piece of work.

With best regards

Morgan Jones

Reviewer #3 (Remarks to the Author):

In my original review of the manuscript entitled "Initial Siberian Traps sill emplacement as the trigger of the end-Permian mass extinction" by Burgess and colleagues, I raised concerns related to the novelty of the argument presented, and of the data to support it. As this fairly harsh assessment was a dissenting view compared to that of other reviewers, it could have been easily dismissed by the authors.

I am thrilled to see the authors took my concerns seriously in their revisions of the manuscript. The novelty of their model is highlighted much more clearly in the revised version of their ms., and they have made significant improvements both in terms of the paper's structure, and formulation of their argument. With the exception of minor typos and editing errors (for instance, the way references are cited on line 79 and elsewhere, in parentheses instead of superscripts; or the absence of a closing parenthesis at the

end of line 128), I would recommend the manuscript be accepted without further revisions.

We have addressed the typos in the manuscript, which now conforms to *Nature* style.